# Attitudes of nursing degree students towards end of life processes. A cultural approach (Spain-Senegal)

E. Begoña García-Navarro[1,2], Miriam Araujo-Hernández[1]*, Alina Rigabert[3], María Jesús Rojas-Ocaña[1]

1 Department of Nursing and Health Sciences, University of Huelva, Huelva, Spain, 2 Research Group ESEIS, Social Studies and Social Intervention, Center for Research in Contemporary Thought and Innovation for Development (COIDESO), University of Huelva, Huelva, Spain, 3 Methodology and Data Analysis Department, Andalusia Beturia Foundation for Health Research (FABIS), Huelva, Spain

These authors contributed equally to this work.

* miriam.araujo@denf.uhu.es

**Data Availability Statement:** All relevant data are within the paper and its Supporting Information files.

## Abstract

### Introduction

The concept of death is abstract, complex and has a number of meanings. Thus, its understanding and the approach taken to it depend, to a large extent, on aspects such as age, culture, training and religion. Nursing students have regular contact with the process of death and so it is of great interest to understand the attitudes they have towards it. As we live in a plural society it is even more interesting to not only understand the attitudes of Spanish students but, also, those of students coming from other countries. In the present article, we seek to identify and compare the attitudes held by nursing degree students at Hekima-Santé University (Senegal) and the University of Huelva (Spain) about end of life processes. The study identifies elements that condition attitudes and coping with death, whilst considering curricular differences with regards to specific end of life training.

### Method

A descriptive, cross-sectional and multi-center study was conducted. The overall sample (N = 142) was divided into groups: Hekima-Santé University (Dakar, Senegal) and the University of Huelva (Huelva, Spain). The measurement instruments used were an ad-hoc questionnaire and Bugen´s Coping with Death Scale.

### Results

Statistically significant differences (p = 0.005, 95%CI) were found in relation to overall Bugen Scale scores. We can confirm that specialized end of life training (University of Huelva, Spain) did not lead to better coping when compared with a population whose academic curriculum did not provide specific training and who engaged in more religious practices (Hekima-Santé University, Senegal).

**Funding:** The author(s) received no specific funding for this work.

**Competing interests:** The authors have declared that no competing interests exist.

## Conclusions

In cultures where religion not only influences the spiritual dimension of the individual, but acts in the ethical and moral system and consequently in the economic, educational and family sphere, the accompaniment at the end of life transcends the formative plane. Considering the plural society in which we live, the training that integrates the Degree in Nursing with regard to the care of the final process, must be multidimensional in which spirituality and faith are integrated, working emotional and attentional skills, as well as cultural competence strategies in this process.

## Introduction

When a person dies, a structural transformation is unfurled within the setting. This has repercussions on the social relationships that existed in their world between relatives, friends and acquaintances, all of which now require readjustment [1]. This restructuring is related to the way in which each culture understands death and the life paradigm it ascribes to [1, 2]. At this time of exceptional health crisis that has caused the pandemic, during which isolation prevails above other aspects, end of life processes are lived out in isolation, without goodbyes or rituals of accompaniment regardless of cultural components. The key axis here is the accompaniment performed by health professionals at the end of life. Despite this, training needs in this area are not regulated by the academic curriculum [3].

All cultures include beliefs, values and customs inherent to the region which have traditionally shaped the meaning of death and the care offered to the terminally ill [4] Thus, it appears to be critical to consider this aspect from a sensitive and respectful perspective with each individual patient and family. For example, in the Western World, it is more likely to conceive the phenomenon of death through the prism of secularism. In this way, death may be tinged with depression, anxiety and sadness, or seen as something natural, as a function of each individuals' life biography and philosophy [5]. This secularism coexists with other religions which that have different interpretation of suffering, death and mourning. This heterogeneity of cultures and beliefs in relation to end of life, often ends up provoking cultural clashes between professionals and patients due to a lack of knowledge about some cultural characteristics [6–8].

Diverse studies highlighted that nursing professionals and students present difficulties inter-relating and communicating with terminally ill patients and their families, whilst also facing issues coping with the death of their patients [9–11]. An individual's coping capacity depends on their personal trajectory, and the values and beliefs they have assimilated over the years [12]. Religious activities within individuals' beliefs have been identified as the most frequently used coping strategies by the general population [13]. This phenomenon provides us with a new insight into the palliative care training of current and future professionals. A study from the University of Cambridge highlighted the relationship between intense anxiety levels in students faced with death and worse psychological health. Such anxiety was also related with negative student attitudes towards palliative care. For this reason, it is important to consider student anxiety provoked by death during health education [14]. Formation training in the degree should include the training of abilities to manage the emotions that arise when faced with the suffering and death of patients [15]. Besides specific training on end of life accompaniment, the academic curriculum should address cultural diversity. However, this should not only consider the immigrant population but also the diversity and complexity of individuals. Many individuals do not identify with a specific culture and have instead created

their own life philosophy, understanding death and palliative care in their own way. The availability of culturally competent end of life care, in which the specific needs of each patient and family are recognized, should be a main concern for health services.

Despite its importance, different authors [16–18] suggest that cultural differences can lead to disparities in end of life care, as a result of insufficient communication and a lack of recognition of individual needs. The provision of palliative care to patients from different cultures requires new ways of conceiving and developing end of life care practices.

An effort was made to tackle the challenges of incorporating a culturally competent end of life approach within nursing degrees in the countries of Sub-Saharan Africa that make up the ECOWAS (Economic Community of West African States), in this way training health professionals. From the University of Huelva (Spain), we applied for an International Cooperation Project. The aim of this was to identify the way in which the academic curriculum approached end of life at, in this case, a university in Dakar, Senegal. Interest in these approaches has increased due to the current health crisis brought about by COVID19, given that the number of deaths has increased as a direct result of the pandemic. Various articles [19–21] outline the importance of examining the preparation and vulnerability of African countries in relation to the pandemic. Despite the priority being to mitigate the biological effects of this illness, we must not forget its psychosocial impact on the population. This impact is not only felt through the social distancing associated with this illness but, also, for the processes of isolation experienced by patients suffering it. This has meant that health professionals now play the leading role of accompanying patients' during their final days and has exposed the lack of information available in relation to this process [22].

In agreement with the arguments presented above, it is evident that in order to lend as much attention as possible to the end of life, specific training is needed on accompaniment, mourning and palliative care. Further, training plans exist at the university stage which aim to work on the skills needed and, specifically, to cope with the process of dying [23]. At the University of Huelva, Spain, this training is integrated within an optional module delivered during the final year Nursing degree with 6 ECTS which includes training in Coping in the end of life, culture, bereavement and compassion. In contrast, at Hekima-Santé University, Senegal, and other universities making up the ECOWAS, such training is not included in the academic curriculum of nursing degrees. In this context, examination of the attitudes of nursing students towards death is fundamental for identifying determining factors and uncovering whether specific training impacts upon student attitudes towards death.

The present article describes the personal and training experiences of nursing degree students attending Hekima-Santé University (Senegal), alongside their attitudes towards the end of life process. Outcomes are compared with the host university of the project, identifying elements that condition attitudes and coping with death.

## Materials and methods

This study employed a descriptive, cross-sectional, observational and multicenter design. Data collection was performed using systematic random sampling, stratified according to university and academic year. The study received approval from the ethics committee of the Andalusian government, Spain, PEIBA with the code (0121N-20.EPAL-2020).

### Sample

The study population was formed by two different universities, namely, Hekima-Santé in Dakar, Senegal and the University of Huelva in Spain. Selected participants were final year

students in order to ensure that they had received all academic curricular content (3rd year for students at Hekima Santé and 4th year for those at the University of Huelva).

Data collection took place between the months of February and March 2020. Study inclusion criteria were as follows: 1) Final year university students undertaking the Nursing Degree at either of the two participating universities and, in the case of University of Huelva participants, who had completed the coping with death module; 2) students who voluntarily agreed to participate in the study and provided written informed consent, and; 3) students who regularly attended classes.

## Procedure

University of Huelva students were administered the questionnaire by one of the project researchers once they had completed the relevant module and signed informed consent. Questionnaires were completed in the student's regular lecture hall at the university. The same process was carried out with the students of Hekima Santé, the director of the institution facilitated us communication with the students to be able to inform about the objectives of the study and the willfulness of participation.

## Instrument

Sociodemographic questions involved two dimensions, intercultural contact, which included questions regarding the culture of origin and language; and contact/experience with death where personal and educational experience items were asked, along with the attitude towards the end-of-life process [25, 26] and the religion professed as well as the practice of it [27].

A version of Bugen's Coping with Death Scale [24] that had been translated and validated in Spanish by Schmidt [25] was used. The Spanish version produced a Cronbach alpha reliability coefficient of 0.86. This tool was developed with the aim of measuring improvements in death-related competence following specific training [24]. In this way, the scale is useful for estimating the benefits of pointed education about death, monitoring whether a seminar on death is effective and, finally, emphasizing that coping is the desirable outcome following educational experiences in relation to death. This scale is a tool composed of 30 items rated along a 7-point Likert scale. A final score is achieved by inverting values for items 1, 13 and 24, and later summing all scores. Scores pertaining to the 33rd percentile or lower indicated inadequate coping, whilst scores pertaining to the 66th percentile or higher indicated optimum coping. All scores in between these two zones highlight adequate or neutral coping. This scale, therefore, enables us to target the abilities required for working in palliative care and evaluates competence in relation to working with end of life patients. The present study used the validated Spanish version of the scale [25] with University of Huelva students and the French version [26] with Hekima-Santé University students.

In order to achieve the proposed objective of considering the two study populations (Senegal-Spain), a number of items were added which related to sociodemographic, religious and cultural characteristics. These questions covered the ethnic differences of those surveyed. Further, some variables were included that related to attitudes towards final processes (experience of death and leave-taking rituals). We used the scale conceived by Núñez-Alarcón [27] in order to capture religious orientations. This scale has a Cronbach alpha reliability coefficient of 0.82.

## Data analysis

Data were analyzed using the software IBM® SPSS® Statistics V24.0.0. Participants sociodemographic variables including sex, age, education, religion and ethnicity were described using

frequencies, percentages, medians and interquartile ranges. Variables related with the phenomenon of death were also described. Frequencies, percentages, medians and interquartile ranges were described. Normality was evaluated via the Kolmogorov-Smirnov test. With the aim examining group differences, the non-parametric Mann-Whitney U test (for independent samples) and chi-squared test (for independent qualitative samples) were employed. In all cases, outcomes were calculated alongside their 95% confidence intervals.

**Ethics procedures.** The study received the corresponding approval from the ethics committee of the Andalusian government, Spain, PEIBA, with the code (0121N-20. EPAL-2020), in addition, this study is part of an international cooperation project subsidized by the Andalusian Agency for International Cooperation, the Andalusian Government of Andalusia, Spain, where the agreement was established between the University of Huelva (Spain) and the University of Hekima Santé (Senegal) for its development.

All participants gave their written consent at the beginning of the questionnaire, after having read the presentation of the study and its purpose. Data were processed in accordance with the Spanish Organic Law 15/1999 of 13 December 1999 on the protection of personal data [Boletín Oficial del Estado (Spanish Offical State Gazette) 298 of 14 December 1999].

The study was conducted in compliance with the ethical and legal standards in force (Declaration of Helsinki).

## Results

### General description

A total of 78 university students from Dakar participated, of which 91% were female and 9% were male. From the population coming from the University of Huelva, 65 students were selected, of which 75.4% were female and 24.6% male (Table 1).

One of the main differences between the two study populations is found in the religion professed by participants and the extent to which they practice it. Of Hekima-Santé students, 87.18% follow Islam. On a Likert scale [27] ranging from 1 (doesn't practice) to 10 (practices often), 59% of responses corresponded to higher involvement in religious practice. On the other hand, 84.6% of University of Huelva students identify as being Christian, however, 70.8% of the Spanish population surveyed reported their involvement in religious practice to correspond to values lower than 5.

Another of the most significant and expected differences relates to participants' culture of origin. Hekima-Santé University students were of a wide range of ethnicities, with more than 63% of students coming from different ethnic backgrounds (31% Wolof, 20% Peul and 12% Serer) while the participants from the other university do not have cultural differences by ethnicity.

**Table 1. Sample characteristics.**

| | | Hekima-Santé University(N = 78) | University of Huelva (N = 65) | p-value |
|---|---|---|---|---|
| **Academic year** | | 3rd | 4th | |
| **Sex** | Female | 71 (91.0%) | 49(75.4%) | 0.011 |
| | Male | 7 (9.0%) | 16 (24.6%) | |
| **Age** | | 24 (IQR = 3) | 21 (IQR = 2) | <0.001 |
| **Religion** | Christianity | 55 (11.54%) | 55 (84.62%) | <0.001 |
| | Buddhist | 0 | 1 (1.54) | |
| | Islam | 68 (87.18%) | 0 | |
| | Other | 0 | 9 (13.85%) | |
| | NA | 1(1.28%) | 0 | |

## Experiences and attitudes towards the end of life process

When exploring personal and formative experiences, in addition to attitudes towards end of life processes, we find differences between the perspectives shared by students attending university in Dakar and those attending the University of Huelva. These differences related to student contact with the phenomenon of death and attitudes regarding the final process. With regards to the variable pertaining to contact with the phenomenon of death, this describes the approach of students to this process at both a personal and a professional level. More than 80% of students of both universities (85.9% Hekima and 83.1% Huelva) referred to connecting in a personal way through the death of close relatives, mainly grandparents. Nonetheless, experience with end-of-life patients during placements or practical lessons is virtually non-existent, with only 33.3% of students having had this experience.

Students were asked to describe this experience in only one word but more than 83.3% of Hekima students and 53.8% of those from Huelva did not provide a response. This reveals the difficulty of response experienced by the study population when dealing with death as a multidimensional concept, and the cultural and educational conditioning factors. Those students from the university in Hekima who did opt to define this phenomenon (25 participants) did so through the concepts presented below in Fig 1: painful/complicated (3.8%), impotence (2.6%), anguish/stress (1.3%), sadness/loneliness (2.6%) or unforgettable/sentimental (5.1%). Statements from students from Huelva were as follows: painful/complicated (12.3%), impotence (4.6%), anguish/stress (6.2%), sadness/loneliness (10.8%) and unforgettable/sentimental (7.7%). Further, students from Huelva added the expression 'vocational' (4.6%) as an additional concept.

## Emotional preparation

With regards to the question about feeling emotionally prepared to care for parents or relatives who were living out their final days, 84.6% of the Hekima-Santé population agreed to feeling

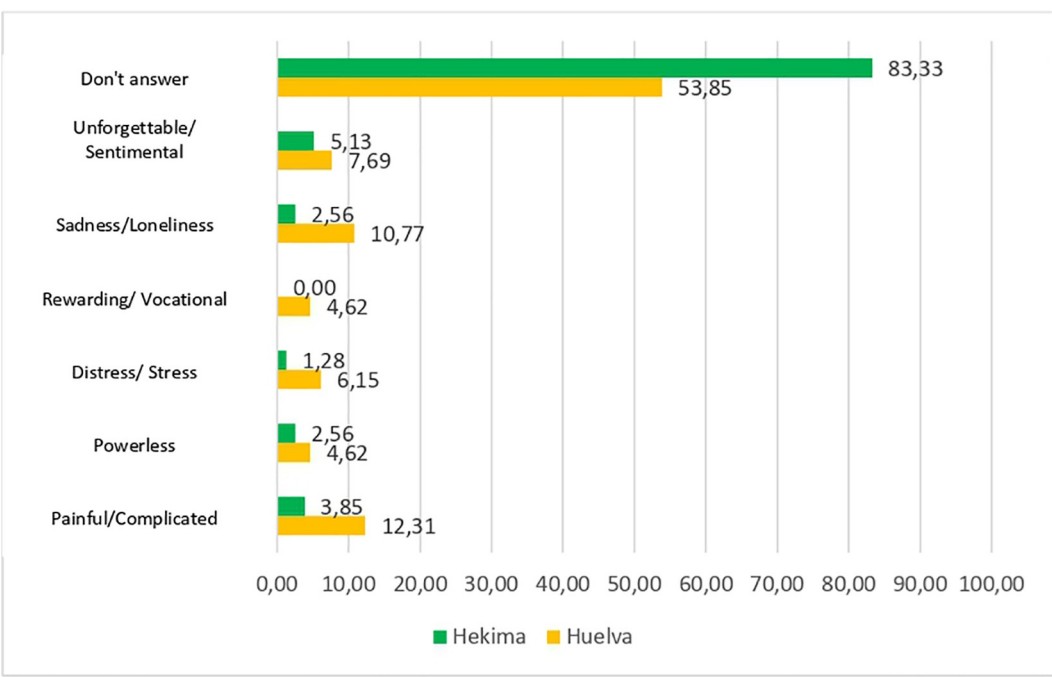

**Fig 1. Definition of the conception of death.** Source: developed by the study authors.

prepared, relative to 60% of University of Huelva. This is despite the fact that the latter are specifically trained to accompany patients at the end of life. We found contrasting outcomes when we asked students about their competence for accompaniment at the end of life, knowledge, and relevant skills and abilities for professionally approaching end of life. Hekima-Santé University students indicated that the most appropriate resources were empathy, understanding and knowing how to accompany (26.9%). In contrast, training on the topic was identified as one of the least highly rated components (11.5%). University of Huelva students similarly gave the greatest value to empathy, understanding and accompaniment (33.8%). However, in contrast to Hekima students, they gave training on the topic the second highest score (20%). This is likely because this aspect is integrated into the academic curriculum at this university.

## Biological process

With regards to opinions of those surveyed in relation to the biological-spiritual tandem of the process of death, 46.2% of Hekima students think that death is a purely biological experience, relative to 16.7% who consider it to be somewhat spiritual. The remaining 37.2% preferred not to respond to this question. Along similar lines, an almost even split was found between those who were afraid of the death of a relative (41.0%) or of their own death (44.9%) and those who were not. In contrast, 84.6% of students from the University of Huelva stated that death was purely biological, with only two individuals deciding not to select one of these dichotomous concepts (biological-spiritual). In the case of these students, they reported greater fear of the death of a loved one (76.9%) than of their own death, with the latter only being reported by a total of 10.8% of the population.

## Cultural influence

In consideration of the cultural impact of the process of death, 41% of the population from Hekima defined a dignified death as the absence of spiritual suffering and having a clear conscience. It is highlighted that 30.8% chose not to respond. A greater variety of concepts were provided by University of Huelva students when arriving to this definition. Concepts included the absence of physical pain, provision of palliative care (24.6%), respecting patient decisions and wishes (24.6%) and being able to accompany in a pleasant setting (21.5%). These outcomes demonstrate greater knowledge about the process.

## Bugen's results

The results associated with the evaluation of the attitudes towards death of our students using the Bugen scale, allowed us to know those students who scored below the 33rd percentile indicate inadequate coping, while those who scored above the 66th percentile correspond to optimal coping. Finally, those who are in the zone between these two percentiles present adequate or neutral coping).

When comparing the general scores on the Bugen Coping with Death Scale, the Huelva students obtained a median of 136 (IR = 34), in relation to a median of 145 (IR = 21) in the Hekima students. This difference was statistically significant (Mann-Whitney U = 1755.00, p = 0.003.

The distribution of reported scores in both participating universities is shown in Table 2.

We can conclude that specialized training in end of life processes (University of Huelva, Spain) does not determine better coping when compared with a population whose academic curriculum does not cover such training but who engage in more religious practices (Hekima-Santé University, Senegal).

**Table 2. Distribution of scores pertaining to Bugen's Coping with Death Scale.**

|  | University of Huelva | Hekima-Santé University |
|---|---|---|
| Inadequate coping | 8 (12.50%) | 1 (1.30%) |
| Adequate coping | 47 (73.44%) | 59 (76.62%) |
| Optimal coping | 9 (14.06%) | 17 (22.08%) |

Table 3 presents Mann-Whitney U comparisons between University of Huelva and Hekima-Santé University students for all Bugen's Coping with Death Scale items.

We obtained (t (141) = p < 0.000) for the following items: I feel prepared to face my death (BS8); I feel prepared to face dying (BS9); Lately I find it OK to think about death (BS12); I feel able to handle the death of others close to me (BS21); I know how to speak to children about death (EB23); I can help someone with their thoughts and feelings about death and dying (EB26), and; I can lessen the anxiety of those around me when the topic is death and dying (BS28). These outcomes point to highly significant differences between the perceptions of students from Senegal and Spain. They indicate that the two groups of students use different strategies, beliefs and attitudes to face death. Students from Dakar show a strong tendency towards higher scores when compared with students from Huelva, despite the latter receiving specialized training. When we observe the distribution of responses to these items, we see that differences are explained by the fact that Hekima-Santé University students were more likely to report total agreement with statements, whereas students from Huelva tended to provide more neutral responses. We can deduct from this behavior that beliefs and religious culture are more important than training at the time of fully defining responses. This is the case even when students are not aware of it given that the majority of students from Hekima did not consider religion to condition their experience of the process (57.7%). Nonetheless, students

**Table 3. P-values pertaining to median comparisons of items found to differ significantly between Huelva and Hekima.**

| Bugen's Coping with Death Scale items (BS) | Median (IQR) Hekima (N = 77) | Median (IQR) Huelva (N = 65) | U de Mann-Whitney | p-value |
|---|---|---|---|---|
| 4. I am aware of the full array of services from funeral homes. | 4(0) | 3(2,75) | 1569,5 | 0.000 |
| 6. I am aware of the full array of emotions which characterize human grief. | 6(3) | 5(3) | 1661 | 0.001 |
| 8. I feel prepared to face my death. | 5(3,25) | 3(2) | 1472 | 0.000 |
| 9. I feel prepared to face my dying process. | 4(3) | 3(2) | 1519 | 0.000 |
| 10. I understand my death-related fears. | 6(2,25) | 5(2) | 1741 | 0.001 |
| 11. I am familiar with prearrangement of funerals. | 4(3,25) | 2(2) | 1790,5 | 0.003 |
| 12. Lately I find it OK to think about death. | 7(2) | 4(3) | 1182 | 0.000 |
| 13. My attitude about living has recently changed. | 6(4) | 4(3,75) | 1909,5 | 0.020 |
| 16. I am making the best of my present life. | 7(1) | 6(1,75) | 1939,5 | 0.022 |
| 20. I will be able to cope with future losses. | 5(3) | 4(2,75) | 1997,5 | 0.047 |
| 21. I feel able to handle the death of others close to me. | 6(3,25) | 4(3) | 1543,5 | 0.000 |
| 22. I know how to listen to others, including the terminally ill. | 6(2) | 5(2) | 1705,5 | 0.001 |
| 23. I know how to speak to children about death. | 6(3) | 3(2) | 813 | 0.000 |
| 24. I may say the wrong thing when I am with someone mourning. | 4(5) | 3(2) | 1802,5 | 0.008 |
| 26. I can help someone with their thoughts and feelings about death and dying. | 6(2) | 5(2) | 1397 | 0.000 |
| 28. I can lessen the anxiety of those around me when the topic is death and dying. | 6(2) | 5(2) | 1392,5 | 0.000 |
| 30. I can tell someone, before I or they die, how much I love them. | 7(1) | 6(2) | 1766 | 0.001 |
| OVERALL SCORE | 145(19,75) | 136(33,75) | 1755 | 0.003 |

from Huelva held the opposite opinion, with 89.2% stating that religion directly influenced the experience of dying.

## Discussion

Diverse studies examining the processes of dying and death [10, 13, 28–31] uncover the cultural, social and religious issues faced by health sciences students who care for and accompany patients and relatives during this process [28–31]. The present study identifies the attitudes towards end-of-life care held by students attending two universities characterized by different cultures, religions, and curricular content different from the death phenomenon.

As described in Table 1, the main differentiating element pertaining to our two populations is that of culture, religious practice and how these dimensions are reflected in the attitude of students.

The main differentiating element pertaining to our two populations is that of culture which is conditioned, at the same time, by the religion followed by each individual and the way in which this impacts upon their attitudes.

Accompaniment at end of life and at death itself differs between the two studied populations as it is influenced by culture and religion. In both Muslim and Christian culture, health is interpreted from a religious standpoint as a right given by God [32]. In Muslim culture–more prevalent amongst Hekima Santé University students (87.18%)–Allah is responsible and it is He who directs the body and life and, consequently, in charge of creating health. Thus, Muslims must be grateful for their health as it has been given by God [33]. In contrast, in Christian culture–predominant in the study population from the University of Huelva (84,62%)–the idea of a predestined life and fulfilling this conception in the future is much less present. Instead, individual responsibility and personal decisions are considered to be what really determine health, as opposed to religion. This coincides with that discussed in a study conducted at the University of the Basque Country [34]. This prior study found that students' attitudes and religious beliefs regarding health, sickness and dying were based less and less on their religious beliefs. These authors explained this as a reflection of the transformation of Spanish society in which religious aspects have been relegated to the background in the social sphere, being largely related with religious holidays. This explains (Table 3) why a higher percentage of Hekima-Santé University students reported coping with death adequately or optimally than did students from the University of Huelva (22,8%). Despite the latter receiving more specific training, we cannot forget that the students of the African University come from a culture where religion is key to give meaning and extrapolate them to educational, economic and family dimensions, but this did not convert into skills which were more objective in nature. For instance, these students were less able to define the concept of death and, when they did, they employed concepts associated with pain and suffering. The dimensions of pain and suffering indicate death-related attitudes and coping which are far from optimal.

As a final point, we emphasize that studies with university students that have used the same tool to evaluate attitudes towards death [24] but have not examined culture as a main focus [34–36], evidenced the need to include specific end-of-life up-skilling within the curricula of health professionals, with training being the only key to improving coping. Nonetheless, studies carried out within different cultural contexts [28, 29, 31], such as was the case in the present study, have found this dimension to be a main axis. An important nuance was highlighted by a study carried out by Pérez de la Cruz [30] within two culturally different populations (Bolivia-Spain) who shared the same religion. This previous study did not observe statistically significant differences between Spanish and Bolivian students (t (540) p = 0.804) according to Bugen's Coping with Death Scale. Their findings, therefore, suggested that differences did not

exist between the perceptions of both groups and that both groups used similar strategies to cope with death. Bolivian students presented a tendency towards higher scores and authors associated this with their greater engagement in religious practices, which enabled them to have a more familiar attitude towards dying. In consideration of our results and reviewed studies [28, 36], students should not only receive instruction in scientific knowledge and technical, but also, should include in their formation the spiritual, cultural and religious dimension. This would enable them to respond appropriately when faced with dying [37].

## Conclusion

Observed differences between the two analyzed samples represent cultural and religious differences surrounding death and the way in which this influences attitudes and care in relation to this phenomenon. Such differences determine the prevailing knowledge around the meaning of death within each culture and community, regardless of whether or not specific training is provided. Especially when we consider cultures where religion directs not only the spiritual dimension of the individual, but their economic, educational and family sphere.

In consideration of the exceptional crisis we are currently living in which we have become more aware of the expiration of human beings, it has become necessary to increase the provision of training programs for nursing students which do not only work at a cognitive level but, also, on their attitudes and cultural strategies that include spirituality and religion. We are a plural society with an obvious need for end of life accompaniment and, for this reason, health professionals must be prepared to provide culturally competent care. The present study has produced evidence that religious culture prevails in attitudes towards death. For this reason, we should broaden research into other cultures with different cultural beliefs.

## Supporting information

**S1 File.**
(SAV)

## Acknowledgments

We thank the University of Hekima Santé, Dakar, Senegal, its faculty and students and its director Souleymane Aliou Diallo for signing the agreement between the two universities subject to studio.

## Author Contributions

**Formal analysis:** Miriam Araujo-Hernández, Alina Rigabert.

**Investigation:** E. Begoña García-Navarro, Miriam Araujo-Hernández, María Jesús Rojas-Ocaña.

**Methodology:** E. Begoña García-Navarro, Alina Rigabert.

**Supervision:** María Jesús Rojas-Ocaña.

**Writing – original draft:** Miriam Araujo-Hernández, María Jesús Rojas-Ocaña.

**Writing – review & editing:** E. Begoña García-Navarro, Miriam Araujo-Hernández.

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
