## [Decision Letter · Decision Letter 0]

6 May 2021

PONE-D-21-06256

Attitudes of nursing degree students towards end of life processes. A cultural approach (Spain-Senegal)

PLOS ONE

Dear Dr. Araujo Hernández,

Thank you for submitting your manuscript to PLOS ONE. After careful consideration, we feel that it has merit but does not fully meet PLOS ONE’s publication criteria as it currently stands. Therefore, we invite you to submit a revised version of the manuscript that addresses the points raised during the review process.

We look forward to receiving your revised manuscript.

Kind regards,

Tareq Mukattash

Academic Editor

PLOS ONE

Journal Requirements:

2. During our internal checks, the in-house editorial staff noted that you conducted research or obtained samples in another country (Senegal). Please check the relevant national regulations and laws applying to foreign researchers and state whether you obtained the required permits and approvals. Please address this in your ethics statement in both the manuscript and submission information. In addition, please ensure that you have suitably acknowledged the contributions of any local collaborators involved in this work in your authorship list and/or Acknowledgements. Authorship criteria is based on the International Committee of Medical Journal Editors (ICMJE) Uniform Requirements for Manuscripts Submitted to Biomedical Journals - for further information please see here: "" ext-link-type="uri" xlink:type="simple">https://journals.plos.org/plosone/s/authorship.""

4. We note you have included a table to which you do not refer in the text of your manuscript. Please ensure that you refer to Table 1 in your text; if accepted, production will need this reference to link the reader to the Table.

Reviewers' comments:

Reviewer's Responses to Questions

**Comments to the Author**

1. Is the manuscript technically sound, and do the data support the conclusions?

Reviewer #1: Partly

Reviewer #2: Yes

Reviewer #3: Partly

2. Has the statistical analysis been performed appropriately and rigorously? 

Reviewer #1: Yes

Reviewer #2: Yes

Reviewer #3: I Don't Know

3. Have the authors made all data underlying the findings in their manuscript fully available?

Reviewer #1: Yes

Reviewer #2: Yes

Reviewer #3: Yes

4. Is the manuscript presented in an intelligible fashion and written in standard English?

Reviewer #1: Yes

Reviewer #2: Yes

Reviewer #3: Yes

5. Review Comments to the Author

Reviewer #1: 1 end-of-life

42 Abstract: conclusion: if you did not find that specialized end-of-life training did not lead to better � how come you conclude that it is necessary to include training within nursing curriculum? (the conclusion needs revision)

52: COVID-19 pandemic?

124 how did you calculate the sample size?

132 this is not an inclusion criterion

133 how did you measure “regularly attend classes”?

135 was there a language barrier to understand the questionnaire?

165 + 166 analyzed ?? what statistical tests were used?

175 why p values here?

185 another (remove of the most) significant and …

188 add: whereas participants from the other university were …

201 any other possible explanations?

241-244 repetition

301 not clear? How come

Reviewer #2: Introduction

I enjoyed reading the introduction and I congratulate the authors for delivering it in such a nice way. However, I have few comments:

- There is no clue in the abstract that the authors are examining nursing students’ attitudes towards death and end of life during COVID-19. I think highlighting this in the abstract is important.

- It would be beneficial to talk more about nursing students’ preparation towards end-of-life aspects in the University of Huelva (Spain). What the end of life curriculum at the university entails?

- I would suggest to clarify how the Spanish culture compared to that of Senegal view death and what are the cultural attitudes towards death in both?

Methods

- The authors mentioned that the design is observational, and this entails performing some form of observation to students although there is no indication that students’ observation was performed!! Please revise the design of the study to reflect the actual participants’ selection and data collection

- What is meant by multicenter design? Selecting nursing students to participate in the study does not apply to the definition of multicenter design

- The authors mentioned that “Data collection was performed using systematic random sampling, stratified according to university and academic year.” At the same time in the description of the sample, it is obvious that the sample was convenient which is different than systematic random sampling , please revise.

- How the researchers obtained the names of students who regularly attended classes?

- more clear and specific description of the instrument dimensions is needed and to give an item example of each dimension.

Results

- What NR in Table 1 stands for?

- Need to cite the tables in text

Discussion

- Well-written and enjoyed reading it.

- Any recommendations for future research?

-I would suggest to have a section that discuss the limitations of the study.

Overall, Nice work!!

Reviewer #3: The topic of end-of-life processes is relevant to the current situation, with the number of deaths exacerbating due to COVID. Additionally, the paper adds to the international knowledge on cultural awareness/competency on the topic. However, there is an great need for justifying the study design and work on the consistency among the research steps, especially, the purpose, methodology and the analysis.

Abstract

Page 2, Lines 39 did not???

Page 2, Lines 38-41 The statement is judgmental. Did you measure religiosity among the two groups?

“We can confirm that specialized end of life training (University of Huelva, Spain) did not lead to better coping when compared with a population whose academic curriculum did not provide specific training and who engaged in more religious practices (Hekima-Santé University, Senegal).”

Introduction

page 3, line 50 change “related with” to “related to”.

page 3, line 61 please avoid generalization without references. “in the Western World, the phenomenon of death is seen through a secular lens and considered to be a merely biological process.”

Page 3, line 64, use plural for “a different interpretation”

Page 3, line 67, “Diverse studies9,10,11 highlight that nursing professionals and students present difficulties inter...”, change the citation place to the end of the statement that refers to the cited work. Also, use a past tense “highlighted”.

Page 4, line 74 same comment for intext citation place. Also, rewrite the sentence “A study from the University of Cambridge…” to clarify the meaning.

Page 4, lines 75-76 add the reference.

Page 4, lines 77-79, what do you mean by “Formation prior to initiating degree studies should include the training of abilities…”

Page 5, line 103, “patients’ ” should be changed to “patient”, no apostrophe.

Page 5, lines 117-118, “Outcomes are compared with the host university of the project, identifying elements that condition attitudes and coping with death.” And page 5, lines 108-114

I am not sure how you are going to compare between the 2 groups in terms of how taking the course actually affect the outcomes. You need to prove that the results are an outcome of the training not due to other variables such as the duration of the program, the number and content of courses they offer in the program, or other factors.

If your aim is to see the effect of the training a pretest-post test design is the one that measures it, but hence you compared between 2 groups from different countries, I doubt that the findings will be owed to the training per se.

Materials and Methods

Page 5, lines 120-122, “Data collection was performed using systematic random sampling, stratified according to university and academic year.”

The sampling technique is not clearly described. Add details about how stratification was performed? What was the number of your target population? How was the randomization performed? Also, “stratified according to university and academic year”, is not consistent with the results that showed only one University from each country and one academic year from each program which appears to reflect convenience sampling.

Sample

Page 6, lines 133-134, the inclusion criteria included “students who regularly 134 attended classes.” What was your parameter for regular attendance? Do you have an attendance rate cut-off point, for example?

Instrument

Page 6-7, lines 145-148, again the statements point to the effectiveness of the training, which does not align with descriptive design.

There is a need for justifying the current study design. And how comparing two distinct groups will serve the purpose of showing the benefit of the training.

Results

Page 8, the findings showed in table 1 were statistically significant which means that the two groups are not similar, how did the study control for these differences?

Page 8, Table 1, the numbers in brackets were describing percentages, but in age they were different. If you mean the standard deviation, please include (SD=2) or (SD=3) for clarity.

Page 9, lines 199-200, did you use open ended questions or qualitative measures? If so please mention them in the instruments and study design parts.

Page 9, lines 201-203 “Those students 202 from the university in Hekima who did opt to define this phenomenon”, please add total number of respondents.

Page 9, lines 210, give the figure a title that descries content.

Pages 10-14 need to be more organized according to the study variables. The use of subheading can be useful.

Table 3, add the result of the statistical tests, not only the significance.

Discussion

Page 14, lines 275-276, not clear.

Page 14, lines 277-279, What findings support this statement? “The main differentiating element pertaining to our two populations is that of culture which is conditioned, at the same time, by the religion followed by each individual and the way in which this impacts upon their attitudes.”

A general comment on the discussion section is that the discussion should flow from the statistically significant findings of the current study. Please link the findings to the discussion and compare or related to findings from previous studies.

Conclusion

Page 16, lines 329-330, I am not sure that the study supports this statement. Please be more consistence to weather culture/religion or training made the differences in outcomes.

6. PLOS authors have the option to publish the peer review history of their article (what does this mean?). If published, this will include your full peer review and any attached files.

Reviewer #1: **Yes: **Aaliyah M Momani

Reviewer #2: No

Reviewer #3: No

---

## [Author Response · Author response to Decision Letter 0]

25 Jun 2021

We would like to thank the Editors and Reviewers for careful and thorough reading of this manuscript and for the thoughtful comments and constructive suggestions, which help to improve the quality of this manuscript. Our response follows. 

Editor’s comments to Author: Responses in blue

https://journals.plos.org/plosone/s/file?id=wjVg/PLOSOne_formatting_sample_main_body.pdfand

We have made the style changes to adapt our proposal to Plos One standards. We have changed the font of the manuscript titles and the format of the bibliography citation. We have also improved the affiliation of the authors, following the recommendations of the journal.

2. During our internal checks, the in-house editorial staff noted that you conducted research or obtained samples in another country (Senegal). Please check the relevant national regulations and laws applying to foreign researchers and state whether you obtained the required permits and approvals. Please address this in your ethics statement in both the manuscript and submission information. In addition, please ensure that you have suitably acknowledged the contributions of any local collaborators involved in this work in your authorship list and/or Acknowledgements. Authorship criteria is based on the International Committee of Medical Journal Editors (ICMJE) Uniform Requirements for Manuscripts Submitted to Biomedical Journals - for further information please see here: https://journals.plos.org/plosone/s/authorship.""

The data collection of our study was developed in different research rooms that we have carried out in Dakar thanks to a cooperation project that has been subsidized by the Andalusian Agency for International Cooperation of the Andalusian Government. For this project to be approved, it was necessary for the University of Huelva and the University of Hekima Santé to sign a prior agreement detailing the research and cooperation project.

We have included this information in the manuscript's ethical statement.

To achieve the proposed objective of describing the personal and training experiences of nursing degree students attending Hekima-Santé University (Senegal) and students from Huelva University (Spain), a number of items were asked in which it was explored sociodemographic, religious, and cultural characteristics. Those items have been included in the document as follows: “Sociodemographic questions involved two dimensions, intercultural contact, which included questions regarding the culture of origin and language; and contact/experience with death where personal and educational experience items were asked, along with the attitude towards the end-of-life process [25,26] and the religion professed as well as the practice of it [27].”

4. We note you have included a table to which you do not refer in the text of your manuscript. Please ensure that you refer to Table 1 in your text; if accepted, production will need this reference to link the reader to the Table.

It was included the reference of table 1 in the Result section, directly after the paragraph in which they are cited for the first time (page 10). 

Reviewers’ comments to Author: Responses in blue

Reviewer #1: 1 end-of-life

42 Abstract: conclusion: if you did not find that specialized end-of-life training did not lead to better ◊ how come you conclude that it is necessary to include training within nursing curriculum? (the conclusion needs revision)

We appreciate your very relevant comment. Thanks to your appreciation we have adhered to a more subjective perception of our conclusions, so we have delved deeper by explaining that formation must be multidimensional in order to meet spiritual and religious cultural diversity. 

52: COVID-19 pandemic?

We have nuanced in the manuscript, detailing the exceptional health crisis generated by the pandemic and how it is very important for the study of the subject that is being investigated.

124 how did you calculate the sample size?

We calculated the necessary sample size based on previous studies [28]. To achieve a power of 80.00% to detect differences in the contrast of the null hypothesis H₀: μ1 = μ2 by means of a bilateral Student's T-test with the Satterthwaite correction for two independent samples, considering that the level of significance is of 5.00%, and assuming that the mean of one group is 49.37 units, the mean of the other group is 55.55 units, the standard deviation of the first group is 15.64 units and that the standard deviation of the second group is 12.20 units, it will be necessary to include at least 72 experimental units in each groups, totaling 144 experimental units in the study. 

132 this is not an inclusion criterion

As it is mentioned in the paper, it was essential for our research that students could ensure a good academic performance according to their level and it was considered their attendance another indicator of great interest and high level. Students who did not attend classes regularly were discarded out since it was probable that they had not reached the level of knowledge it was required for this research. 

133 how did you measure “regularly attend classes”?

Thank you for your appreciation, this has led us to improve the explanation in the text. We proceed to describe the explanation for this question. Although the sample is random, class attendance was an inclusion criterion. That is to say, the regularity in the attendance of the students was verified before the randomization, and later the survey was randomly administered. A clarification has been added in the text to facilitate the reader's understanding, “(minimum attendance of 80%)”.

135 was there a language barrier to understand the questionnaire?

Thank you for your appreciation, we have included in the manuscript, in the section of methodology, procedure, the participation of the director of the institution for the interpretation and effective communication with the students, specifying objectives of study and willfulness of the same.

165 + 166 analyzed ?? what statistical tests were used?

Considering that it may cause confusion the information has been modified into the following sentence: “Participants’ sociodemographic variables including sex, age, education, religion and ethnicity were described using frequencies, percentages, medians and interquartile ranges. Variables related with the phenomenon of death were also described.”

175 why p values here?

It was contemplated that both samples, Hekima and Huelva, might display different distributions in their sociodemographic variables. In order to display their both distributions and describe the sample, significance (p) is shown for the sake of clarity and comparability. This would allow the researcher to establish conclusions regarding the external validity. 

185 another (remove of the most) significant and …

We appreciated the improvement. The text has been modified as suggested. The final sentence is the following: “Another significant and expected difference relates to participants’ culture of origin.”

188 add: whereas participants from the other university were… 

We added this clarification in the text, following your suggestion: “while the participants from the other university do not have cultural differences by ethnicity.” 

201 any other possible explanations? 

We appreciate the suggestion. We have expanded the explanation in the text, being more consistent with the results presented, as follows: “This reveals the difficulty of response experienced by the study population when dealing with death as a multidimensional concept, and the cultural and educational conditioning factors.”

241-244 repetition 

Thank you for your consideration, we have delved into the explanation of both instruments so as not to provide the information.

301 not clear? How come

Hekima Santé students present an adequate optimal coping, however these values are not reflected in a practical way, since their description of death, the elements with which they relate it, do not demonstrate this confrontation. This is due to the influence that culture and religion has on this culture, being an important condition to take into account in its responses and in the way it reflects its confrontation with death.

But given your important appreciation, we have been able to nuance this idea to make it easier for the reader to understand.

I enjoyed reading the introduction and I congratulate the authors for delivering it in such a nice way. However, I have few comments: 

Thank you very much for your words, our team is very flattered and motivated.

- There is no clue in the abstract that the authors are examining nursing students’ attitudes towards death and end of life during COVID-19. I think highlighting this in the abstract is important. 

Thank you very much for your appreciation. Concretely, covid is not the objective of this study, the aim is to know the attitudes of the students of the degree in nursing about the end-of-life process. The pandemic has made visible the absence of nursing training regarding death, the necessity of developing more work in this field, to know what dimensions we must emphasize in the academic curriculum of future professionals in order to accompany effectively.

- It would be beneficial to talk more about nursing students’ preparation towards end-of-life aspects in the University of Huelva (Spain). What the end of life curriculum at the university entails?

Taking into account the timing of your comment, we have included more information regarding the subject that addresses this topic at the University of Huelva. At the University of Huelva, Spain, this training is integrated within an optional module delivered during the final year Nursing degree with 6 ECTS which includes training in Coping in the end of life, culture, bereavement and compassion.

- I would suggest to clarify how the Spanish culture compared to that of Senegal view death and what are the cultural attitudes towards death in both?

Thank you for your suggestion. We have included new information in the conclusions section, clarifying that topic. 

Methods

- The authors mentioned that the design is observational, and this entails performing some form of observation to students although there is no indication that students’ observation was performed!! Please revise the design of the study to reflect the actual participants’ selection and data collection

We appreciate your comment. Clinical studies can be divided into two broad categories: experimental trials, in which the investigator intervenes and observational studies, in which the investigator performs no intervention (Thadhani, 2006). During this study, no experiment was taken. The investigators collected information regarding attitudes towards the end of life process identifying elements that condition attitudes and coping with death.

- What is meant by multicenter design? Selecting nursing students to participate in the study does not apply to the definition of multicenter design.

It is commonly accepted in the research literature to refer to multicenter study as a scientific research that is carried out at more than one institution or center. Multicenter research confers many distinct advantages over single-center studies, including larger sample sizes for more generalizable findings, sharing resources amongst collaborative sites, and promoting networking (Cheng et al., 2017, chung et al., 2010). In this study, we count on two academical institutions located in Huelva (Spain) and Hekima (Senegal), where we collected the data. 

- The authors mentioned that “Data collection was performed using systematic random sampling, stratified according to university and academic year.” At the same time in the description of the sample, it is obvious that the sample was convenient which is different than systematic random sampling, please revise.

The researchers had a list of the total sample of students of each university. With the aim of randomly include students in our research, it was generated a random number list that was associated with the students total list, therefore every student of the total sample had an equal chance of being selected. 

- How the researchers obtained the names of students who regularly attended classes?

Both Universities provided the number of identification of the students attending classes. It was an anonymous list since student’s names or any identification were not shown on that lists. 

- more clear and specific description of the instrument dimensions is needed and to give an item example of each dimension.

Bugen’s coping death scale was originally conceived as a one dimension scale (Robbins, 1990) and the evidence collected by Galiana et al., (2014) offered support for the one-dimensional structure proposed. Additionally, it was considered as a one dimension scale in our study and there is no analysis performed using a multidimensional model associated with Bugen’s scale. 

Results

What NR in Table 1 stands for? 

We apologize and thank you for your appreciation, it was a translation error. It means "no answer", the correct abbreviation is NA.

- Need to cite the tables in text. 

Thank you for your contribution, the table has been cited in the text as indicated.

Discussion

- Well-written and enjoyed reading it. 

Thank you very much again, it is a pleasure to receive that feedback from someone who reads our work in the role of reviewer. We deeply appreciate it, and it gives us a lot of motivation.

- Any recommendations for future research? 

Include the multidimensional component of death in those subjects related to end-of-life care or palliative care, taking into account spirituality, religion and culture.

We must emphasize that it is necessary to include this line in the academic curriculum of nurse discipline.

-I would suggest to have a section that discuss the limitations of the study.

We've included it

Overall, Nice work!! 

Thank you for your time and for your comments, we appreciate the positive feedback from the reviewer.

Reviewer #3: The topic of end-of-life processes is relevant to the current situation, with the number of deaths exacerbating due to COVID. Additionally, the paper adds to the international knowledge on cultural awareness/competency on the topic. However, there is an great need for justifying the study design and work on the consistency among the research steps, especially, the purpose, methodology and the analysis.

Abstract

Page 2, Lines 39 did not??? 

Thank you for your contribution, we have made changes to further clarify this part of the article.

Page 2, Lines 38-41 The statement is judgmental. Did you measure religiosity among the two groups? “We can confirm that specialized end of life training (University of Huelva, Spain) did not lead to better coping when compared with a population whose academic curriculum did not provide specific training and who engaged in more religious practices (Hekima-Santé University, Senegal).”

We appreciate your suggestion. In the summary we have not been able to deepen the results,but in the manuscript we have detailed this nuance, specifically on pages 10-11 or (Table 1. p. 10 -11)

Introduction

page 3, line 50 change “related with” to “related to”. 

Thank you very much for your appreciation, we have made this change to the manuscript.

page 3, line 61 please avoid generalization without references. “in the Western World, the phenomenon of death is seen through a secular lens and considered to be a merely biological process.” 

Thank you again, for your correct appreciation we have been able to see that generalisation does not benefit the reader's understanding, so we have made its modification.

Page 3, line 64, use plural for “a different interpretation” 

Thank you very much, we have modified the manuscript with the expression you propose.

Page 3, line 67, “Diverse studies9,10,11 highlight that nursing professionals and students present difficulties inter...”, change the citation place to the end of the statement that refers to the cited work. Also, use a past tense “highlighted”. 

We have made the change that you propose, understanding that it represents an improvement in the presentation of the manuscript. Thanks for the indication.

Page 4, line 74 same comment for intext citation place. Also, rewrite the sentence “A study from the University of Cambridge…” to clarify the meaning. 

We have made the change suggested, we understand that it represents an improvement in the presentation of the manuscript. Thanks for the indication.

Page 4, lines 75-76 add the reference. 

In making the previously proposed modification, we have drafted the reference in its proper place,justifying the idea of the manuscript.

Page 4, lines 77-79, what do you mean by “Formation prior to initiating degree studies should include the training of abilities…” 

Thank you for your appreciation, we have noticed an error in the concept of "undergraduate", wanting to refer to the undergraduate student in nursing. We have made the modification to the text.

Page 5, line 103, “patients’ ” should be changed to “patient”, no apostrophe. 

 Thank you for your appreciation, it was an error that has been modified in the text.

Page 5, lines 117-118, “Outcomes are compared with the host university of the project, identifying elements that condition attitudes and coping with death.” And page 5, lines 108-114 I am not sure how you are going to compare between the 2 groups in terms of how taking the course actually affect the outcomes. You need to prove that the results are an outcome of the training not due to other variables such as the duration of the program, the number and content of courses they offer in the program, or other factors.

If your aim is to see the effect of the training a pretest-post test design is the one that measures it, but hence you compared between 2 groups from different countries, I doubt that the findings will be owed to the training per se.

Thank you for your interesting suggestion. As described in the manuscript, no experiments were conducted during this study. Researchers gathered information on attitudes towards the end of the life process by identifying elements that condition attitudes and addressing death. One of the groups, the University of Huelva, has a training program, in the case of the Hekima-Sante group that does not have. The intention of the researchers was to illustrate the differences that occur in two societies of different cultures, where in one of them, religion (internal component) influences more external dimensions such as education and training. The importance of these results reveals the need for multidimensional training in the degree of nursing, as we live in plural and contemporary societies where different cultures and religions converge.

Materials and Methods

Page 5, lines 120-122, “Data collection was performed using systematic random sampling, stratified according to university and academic year.”

The sampling technique is not clearly described. Add details about how stratification was performed? What was the number of your target population? How was the randomization performed? Also, “stratified according to university and academic year”, is not consistent with the results that showed only one University from each country and one academic year from each program which appears to reflect convenience sampling.

Sample

- Add details about how stratification was performed.

Thank you very much for your comment, as it was right and it was a mistake. The text has been modified.

- What was the number of your target population?

We calculated the necessary sample size based on previous studies [28]. To achieve a power of 80.00% to detect differences in the contrast of the null hypothesis H₀: μ1 = μ2 by means of a bilateral Student's T-test with the Satterthwaite correction for two independent samples, considering that the level of significance is of 5.00%, and assuming that the mean of one group is 49.37 units, the mean of the other group is 55.55 units, the standard deviation of the first group is 15.64 units and that the standard deviation of the second group is 12.20 units, it will be necessary to include 72 experimental units in each groups, totaling 144 experimental units in the study. 

- How was the randomization performed

The researchers had a list of the total sample of students of each university. With the aim of randomly include students in our research, it was generated a random number list that was associated with the students total list, therefore every student of the total sample had an equal chance of being selected. 

Page 6, lines 133-134, the inclusion criteria included “students who regularly 134 attended classes.” What was your parameter for regular attendance? Do you have an attendance rate cut-off point, for example?

Yes, the attendance criterion is based on having attended a minimum of 80% of presential classes. For clarity in the text, we have included it as a clarification in parentheses.

Instrument

Page 6-7, lines 145-148, again the statements point to the effectiveness of the training, which does not align with descriptive design.

There is a need for justifying the current study design. And how comparing two distinct groups will serve the purpose of showing the benefit of the training.

Thank you for your interesting suggestion. We have stated in the paper “In this way, the scale is useful for estimating the benefits of pointed education about death, monitoring whether a seminar on death is effective and, finally, emphasizing that coping is the desirable outcome following educational experiences in relation to death.” We are now aware that the word “effective” might cause confusion regarding the aim of this study. As it has been stated, our main objective was to describe both populations, Hekima and Huelva. There was no intention of proving any effectiveness of the instrument as it has been already proven. Also, the Bugen’s scale has been used for estimating the benefits of pointed education about death. Therefore, the researchers’s intention was to throw some light into the elements pertaining to our two populations, specifically culture which is conditioned, at the same time, by the religion followed by each individual and the way in which this impacts upon their attitudes

Results

Page 8, the findings showed in table 1 were statistically significant which means that the two groups are not similar, how did the study control for these differences?

Thank you for your appreciation. Undeniably, these two populations have statistical different distributions in their variables. However, it was not in the researchers intentions to control these variables, as the design of the study is merely descriptive, as it has been commented. 

Page 8, Table 1, the numbers in brackets were describing percentages, but in age they were different. If you mean the standard deviation, please include (SD=2) or (SD=3) for clarity.

Thank you for your suggestion, it has been modified in the text. 

Page 9, lines 199-200, did you use open ended questions or qualitative measures? If so please mention them in the instruments and study design parts. 

Thank you for your appreciation, we have modified the manuscript to encourage the reader's understanding. Our instrument, in sociodemographic items, only an open question was included in relation to the dimension "Contact with the phenomenon of death", when asking that the concept of death be described. Subsequently, in the analysis of the data this variable was categorized following the response pattern of the participants.

Page 9, lines 201-203 “Those students 202 from the university in Hekima who did opt to define this phenomenon”, please add total number of respondents.

Thank you for your appreciation. We have included this information in the manuscript.

Page 9, lines 210, give the figure a title that descries content.

Thank you very much for your appreciation, we had not noticed this error. We have already included the title of Figure 1 in the text.

Pages 10-14 need to be more organized according to the study variables. The use of subheading can be useful.

We appreciate your suggestion. It has been implemented in the text different subheadings, described below: 

● General description

● Experiences and attitudes towards the end of life process

● Emotional preparation

● Biological process

● Cultural influence 

● Bugen's results

Table 3, add the result of the statistical tests, not only the significance.

We appreciate this modification, it has improved the quality of this paper.

Discussion

Page 14, lines 275-276, not clear.

Thank you very much for your appreciation, we have warned that the reading might seem confusing, therefore we have modified the text accordingly.

Page 14, lines 277-279, What findings support this statement? “The main differentiating element pertaining to our two populations is that of culture which is conditioned, at the same time, by the religion followed by each individual and the way in which this impacts upon their attitudes.”

A general comment on the discussion section is that the discussion should flow from the statistically significant findings of the current study. Please link the findings to the discussion and compare or related to findings from previous studies.

We appreciate your considerations. We have modified and nuanced the discussion section to respond to your assessment.

Conclusion

Page 16, lines 329-330, I am not sure that the study supports this statement. Please be more consistence to weather culture/religion or training made the differences in outcomes.

Thank you very much for your very correct comment, we have followed your instructions in the discussion which have led to a more coherent conclusion, thank you.

---

## [Decision Letter · Decision Letter 1]

6 Jul 2021

Attitudes of nursing degree students towards end of life processes. A cultural approach (Spain-Senegal)

PONE-D-21-06256R1

Dear Dr. Araujo Hernández,

We’re pleased to inform you that your manuscript has been judged scientifically suitable for publication and will be formally accepted for publication once it meets all outstanding technical requirements.

Kind regards,

Tareq Mukattash

Academic Editor

PLOS ONE

Additional Editor Comments (optional):

Reviewers' comments:

Reviewer's Responses to Questions

**Comments to the Author**

1. If the authors have adequately addressed your comments raised in a previous round of review and you feel that this manuscript is now acceptable for publication, you may indicate that here to bypass the “Comments to the Author” section, enter your conflict of interest statement in the “Confidential to Editor” section, and submit your "Accept" recommendation.

Reviewer #3: All comments have been addressed

2. Is the manuscript technically sound, and do the data support the conclusions?

Reviewer #3: Yes

3. Has the statistical analysis been performed appropriately and rigorously? 

Reviewer #3: Yes

4. Have the authors made all data underlying the findings in their manuscript fully available?

Reviewer #3: Yes

5. Is the manuscript presented in an intelligible fashion and written in standard English?

Reviewer #3: Yes

6. Review Comments to the Author

Reviewer #3: (No Response)

7. PLOS authors have the option to publish the peer review history of their article (what does this mean?). If published, this will include your full peer review and any attached files.

Reviewer #3: No

---

## [Editor Report · Acceptance letter]

19 Jul 2021

PONE-D-21-06256R1 

Attitudes of nursing degree students towards end of life processes. A cultural approach (Spain-Senegal) 

Dear Dr. Araujo Hernández:

I'm pleased to inform you that your manuscript has been deemed suitable for publication in PLOS ONE. Congratulations! Your manuscript is now with our production department. 

Kind regards, 

on behalf of

Dr. Tareq Mukattash 

Academic Editor

PLOS ONE